# Enhancing Model Robustness Against Noisy Labels via Kronecker Product Decomposition

## Abstract

Deep learning models have made remarkable progress across various domains in recent years. These models heavily rely on large-scale datasets for training, and a noisy dataset can degrade the performance of the model. To train accurate deep learning models, it is crucial to develop training algorithms that are robust to noisy training data and outliers while ensuring high performance. In this work, we study the problem of model training under noisy labels/outputs and propose a method based on Kronecker product decomposition to improve robustness during training. The proposed method is easy to implement and can be readily combined with robust loss functions. We report results from experiments conducted on both classification and regression tasks in the presence of noisy labels/outputs. Our results demonstrate that our approach outperforms existing robust loss methods in terms of model performance.

## 1 Introduction

Deep learning has made significant progress across various domains, primarily by learning the distribution of training data to generalize to new data. However, if the training data differs from the test data (e.g., due to noisy labels), model performance can degrade significantly. Thus, preparing a high-quality training dataset is crucial for building an accurate model. Human annotation is precise but costly and tedious, while crowdsourcing is faster and cheaper but introduces more noise. In large-scale datasets, noise is unavoidable. Therefore, a key characteristic of deep learning models is robustness—the ability to maintain strong performance despite noisy or corrupted training data.

Research has addressed the noisy label problem through pre-training, post-training, and in-training. Pre-training techniques, such as data pruning, refining the training dataset by selecting a smaller yet more informative subset, reducing noise while preserving essential learning signals (Park et al., 2023). Post-training techniques, on the other hand, leverage a pre-trained model to identify and remove samples that heavily influence the decision boundary. By eliminating these outliers and fine-tuning the model, generalization performance can be further improved (Park et al., 2021). In-training methods enhance robustness by using robust loss functions (Kim et al., 2021; Li et al., 2020), which mitigate the impact of noisy labels without modifying the training strategy or model architecture.

In this paper, we focus on in-processing techniques to enhance model robustness against noisy labels. Specifically, we leverage Kronecker Product Decomposition (KPD) (Van Loan, 2000) during training. We will demonstrate how to incorporate KPD into model training and provide both theoretical and experimental analyses showing its effectiveness in improving robustness to noisy labels in the training dataset. Additionally, KPD can be combined with various robust loss functions (Ghosh et al., 2017b; Huang et al., 2023; Samuel & Chechik, 2021) to further improve robustness. It is worth noting that KPD can decrease or increase the number of training parameters, depending on the settings used during training. While it is commonly believed that deep learning models with more parameters generalize better, some research suggests that models with fewer parameters can achieve comparable performance to larger models (Frankle & Carbin, 2019). In some cases, smaller models may even generalize better (Nakkiran et al., 2019; Liu et al., 2019;

Van Sieleghem et al., 2022). Therefore, we provide further empirical study on how the parameters of KPD impact model robustness. To summarize, our main contribution in this paper can be listed as follows:

**1.** We demonstrate theoretically that the the ground truth weight matrix trained with normal empirical risk minimization method is not identifiable with noisy data. Replacing each weight matrix with Kronecker Product Decomposition (KPD) or Rank Factorization, which is a special case of KPD, might be able to improve robustness.

**2.** We also show that KPD and rank factorization can be used as a plug-in with other robust techniques, such as robust loss functions. When combined with these loss functions, we observe even greater improvements in model robustness.

**3.** We conduct several experiments on both synthetic and real-world data, covering regression and classification tasks to demonstrate our empirical results are aligned with the theoretical findings. Furthermore, we provide an ablation study to explore the impact of KPD parameters on the model's performance.

The remainder of this paper is organized as follows: In Section 2, we introduce related work. Section 3 covers the notations and preliminaries on KPD and robust loss functions. In Section 4, we discuss our theoretical findings and present our proposed robust training algorithm. Experiment results on synthetic and real-world data are provided in Section 5, and we conclude the paper in Section 6.

## 2 Related work

### 2.1 Robust Regression

Robust regression methods have gained significant attention due to their ability to handle outliers and model deviations from assumptions like normality Jambulapati et al. (2021); Bhatia et al. (2017); Chen & Paschalidis (2018). Classic regression techniques, such as Ordinary Least Squares (OLS)Kuchibhotla et al. (2018), are sensitive to outliers, which can disproportionately affect the estimated coefficients, leading to biased models. In response, robust regression methods aim to mitigate the influence of outliers and heavy-tailed errors by assigning different weights to the observations or using alternative loss functions. Robust Regression problem can be traced back to Huber LossHuber (1992a). For alternative loss functions, Belagiannis et al. (2015) also propose Tukey's biweight function to achieve robustness. And Chen & Paschalidis (2018) uses regularization to make the training more robust to the noise.

### 2.2 Robust Classification with Noisy Labels

As the capabilities of deep learning models continue to improve, the amount of required data is increasing exponentially. High-quality annotations are not only costly but also prone to introduce labeling errors. Different approaches are proposed to handle this problem, mostly can be divided into three categories: design loss criteria, correct noise label and refine training strategies.

Design loss criteria methods would modify the loss function to maintain the robustness. Wang et al. (2019) propose Symmetric Cross Entropy based on the idea of KL-divergence to increase the gradients if the network is more confident about its prediction at the labeled class. Mean Average Error (MAE) loss is proven to contribute to the robustness of the model Ghosh et al. (2017b), however, it can also lead to slower model convergence. Zhang & Sabuncu (2018) adopt the Box-Cox transformation to the MAE loss which would not only retain the robustness characteristic of MAE loss but also adaptively assign higher weights to difficult samples as cross entropy. Xu et al. (2019) propose an information-theoretic robust loss function based Determinant based Mutual Information. This loss function can provide a guarantee of robustness to instance independent label noise, regardless of the noise pattern. In another line of research, Ma et al. (2020) modify loss functions like cross entropy loss and focal loss and propose normalized cross entropy and normalized focal loss. This normalization improves robustness to noisy labels Ghosh et al. (2017a). Another approach to handle noisy labels is to correct noisy labels, typically using semi-supervised learning methods Li et al. (2020); Kim et al. (2021). These methods require determining whether a sample contains noise at the beginning. For instance, Kim et al. (2021) uses the eigenvectors of embeddings within the same class to

make this determination. In addition to the mentioned methods, we can also refine the training strategy, e.g., by adding an additional noise adaptation layer Goldberger & Ben-Reuven (2017).

It is worth mentioning that some robust classification approaches can be adapted to the regression task by discretizing the output Torgo & Gama (1997).

### 2.3 Weight Matrix Decomposition

Matrix decomposition methods such as Singular Value DecompositionLouizos et al. (2017), Low-rank DecompositionHu et al. (2021); Zhang et al. (2015), Tucker DecompositionLebedev et al. (2015), and Kronecker Product Decomposition Tahaei et al. (2021); Ahmadi-Asl et al. (2024) have various applications in deep learning literature. They can be used to decrease the model's parameters Hu et al. (2021), finding interpretable component Praggastis et al. (2022), and noise reduction Buchanan et al. (2018), to name a few. There are several works that try to recover matrices through noisy measurements with a combination of sparse and low-rank factorization Zhou & Tao (2011); Wright et al. (2009). However, there is no work that studies the impact of matrix decomposition on robustness of model training with noisy labels.

## 3 Notations and Preliminaries

### 3.1 Notations

Considering a supervised learning problem of predicting target $y \in \mathcal{Y}$ from observed features $x \in \mathcal{X}$. In the classification task, $\mathcal{Y} = \{0, 1, \cdots, K-1\}$, where $K$ is the number of classes. In regression tasks, $\mathcal{Y} = \mathbb{R}^K$. We denote clean training dataset with $N$ samples as $\mathcal{D}_g = \{(x_i, y_i) | i = 1, \ldots, N\}$. We assume that $y_i$ is not accessible due to noisy observation. Instead, we have access to the observed training dataset denoted by $\mathcal{D} = \{(x_i, \hat{y}_i) | i = 1, \ldots, N\}$, where $\hat{y}_i$ is the noisy label generated by a randomized corruption function $\mathrm{H} : \mathcal{Y} \mapsto \mathcal{Y}$.

We denote an $L-$layer neural network parameterized by $W^{[1]}, ..., W^{[L]}$ by $f(x; W^{[1]}, \cdots, W^{[L]})$. In regression problems, $f(x; W^{[1]}, \cdots, W^{[L]}) \in \mathbb{R}^K$, and in classification problems, $f(x; W^{[1]}, \cdots, W^{[L]}) \in [0, 1]^{|\mathcal{Y}|}$ is a vector of probabilities that identifies the likelihood of each class label $y \in \mathcal{Y}$ given an input $x$. For notational convenience, sometimes we omit $W^{[1]}, \cdots, W^{[L]}$ and use $f(x)$ to denote the predictor and $f_j(x)$ to denote the probability of class $j$ given an input $x$. Given loss function $\mathcal{L}(f(x), \hat{y})$, the empirical risks $\mathcal{R}_\mathcal{L}$ w.r.t. datasets $\mathcal{D}_g = \{(x_i, y_i)\}_{i=1}^N$ and $\mathcal{D} = \{(x_i, \hat{y}_i)\}_{i=1}^N$ are given by,

$$
\begin{aligned}
\mathcal{R}_\mathcal{L}(f; \mathcal{D}_g) &= \frac{1}{N} \sum_{i=1}^N \mathcal{L}(f(x_i; W^{[1]}, \cdots, W^{[L]}), y_i), \\
\mathcal{R}_\mathcal{L}(f; \mathcal{D}) &= \frac{1}{N} \sum_{i=1}^N \mathcal{L}(f(x_i; W^{[1]}, \cdots, W^{[L]}), \hat{y}_i).
\end{aligned}
\tag{1}
$$

### 3.2 Kronecker Product Decomposition

Consider matrices $A \in \mathbb{R}^{m_1 \times n_1}$ and $B \in \mathbb{R}^{m_2 \times n_2}$. The Kronecker product of $A$ and $B$, denoted by $A \otimes B$, is an $m_1 m_2$ by $n_1 n_2$ matrix calculated as follows,

$$
A \otimes B := \begin{bmatrix} a_{1,1} B & \cdots & a_{1,n_1} B \\ \vdots & \ddots & \vdots \\ a_{m_1,1} B & \cdots & a_{m_1,n_1} B \end{bmatrix}
\tag{2}
$$

According to Van Loan (2000), any matrix $W \in \mathbb{R}^{m \times n}$ with $m = m_1 m_2, n = n_1 n_2$ can be decomposed using the Kronecker product as follows,

$$
W = \sum_{i=1}^R A_i \otimes B_i = \sum_{i=1}^R W_i,
\tag{3}
$$

where $A_i \in \mathbb{R}^{m_1 \times n_1}$, $B_i \in \mathbb{R}^{m_2 \times n_2}$, and $R$ is the rank of decomposition. Notably, the **Rank Factorization** is a special case of Kronecker product decomposition where $m_1 = m, n_1 = 1$ and $m_2 = 1, n_2 = n$. Furthermore, after the Kronecker product decomposition of matrix $W$, $Wx$ can be calculated as follows, $Wx = vec(\sum_{i=1}^{R} B_i \check{x} A_i^T)$, where $x \in \mathbb{R}^{n \times 1}$, $\check{x} \in \mathbb{R}^{n_2 \times n_1}$ is generated by reshaping $x$, and $vec(\cdot)$ is an operation for turning a matrix into a vector.

### 3.3 Noise Model

Label noise is a common issue in machine learning. It happens when the labels/target variables on a dataset are wrong or inconsistent, often because of human mistakes or ambiguous data. This issue makes it hard for ML models to find accurate patterns hurting their performance.

In this paper, we consider the corruption function as follows,

$$\hat{y}_i = \mathrm{H}(y_i) = \begin{cases} y_i & \text{with probability } 1 - \eta_i \\ y_i + \mathcal{N}_i & \text{with probability } \eta_i \end{cases}, \tag{4}$$

where $\mathcal{N}_i$ is the noise (e.g., Gaussian noise in a regression tasks, or a discrete noise in a classification task), and $\eta_i$ is the noise ratio.

### 3.4 Robust Loss Functions

While it is common to use Cross Entropy (CE) loss and Mean Squared Error (MSE) loss for training a classifier or regressor, they are not robust to the output/label noise in the training dataset. As a result, the models trained on noisy dataset $\mathcal{D}$ using CE/MSE do not generalize well on unseen data. Using a robust loss function instead of CE or MSE is a common method to make the training process robust to noisy labels/outputs. Various robust loss functions including MAE loss, Huber loss, Tukey loss, and Symmetric/Generalized/Normalized Cross Entropy loss Huber (1992b); Belagiannis et al. (2015); Wang et al. (2019); Zhang & Sabuncu (2018); Xu et al. (2019); Ma et al. (2020) have been introduced in the literature for regression and classification tasks. We will review the robust loss functions used in our experiments in Appendix.

## 4 Robust Training Using Kronecker Product Decomposition

In this section, we examine how Kronecker Product Decomposition (KPD) affects training an ML model with noisy labels. We first present a theoretical result demonstrating that KPD improves robustness to label noise on single linear layer networks. Building on this insight, we then propose a robust training algorithm designed to handle noisy labels.

### 4.1 Theoretical Study

In this part, we consider a linear multi-output regression model where the goal is to estimate matrix $W^* \in \mathbb{R}^{m \times n}$ from noisy observations $\{(x_i, \hat{y}_i) | i = 1, \ldots, N\}$, where $x_i \in \mathbb{R}^m$ are i.i.d standard Gaussian vectors, $y_i \in \mathbb{R}^K$ with $K = n$, and $\hat{y}_i = y_i + \epsilon_i$ with $y_i = W^* x_i$. Moreover, $\epsilon_i$'s are independently drawn from a distribution. In this part, we make an assumption that $\epsilon_i = 0$ with probability $\eta$, and with probability of $1 - \eta$, $\epsilon_i$ follows noise distribution $P_0$. We further assume that if random variable $\mathcal{E}$ follows distribution $P_0$, then $\mathbb{E}_{\mathcal{E} \sim P_0}[\mathcal{E}^{(j)}] = 0$, and there is a non-negative constant $t_0$ and a non-negative constant $p_0$ such that $\Pr\{\mathcal{E}^{(j)} \geq t_0\} \geq p_0$ for all $j$, where $\mathcal{E}^{(j)}$ is the $j$-th entry of $\mathcal{E}$.

Let $(A_j^*, B_j^*)_{j=1}^R$ be the KPD for $W^*$, that is, $W^* = \sum_{j=1}^{R}(A_j^* \otimes B_j^*)$, where $A_j^* \in \mathbb{R}^{m_1 \times n_1}$ and $B_j^* \in \mathbb{R}^{m_2 \times n_2}$. Since KPD and its rank for $W^*$ is not unique, we define set $\mathcal{S}$ as follows,

$$\mathcal{S} = \left\{ (A_j, B_j)_{j=1}^R \middle| \sum_{j=1}^{R}(A_j \otimes B_j) = W^*, R > 0 \right\}.$$

Our goal is to find $W^*$ by solving a robust regression problem. In general, $W^*$ can be found by solving the following problem (note that $l_1$ loss is commonly used in robust machine learning),

$$\mathcal{R}_\mathcal{L}(W; \mathcal{D}) = \frac{1}{N} \sum_{i=1}^N \|W x_i - \hat{y}_i\|_1, \tag{5}$$

$$\hat{W} = \arg\min_W \mathcal{R}_\mathcal{L}(W; \mathcal{D}). \tag{6}$$

On the other hand, if we want to use KPD, we solve the following *non-convex* problem,

$$(\hat{A}_j, \hat{B}_j)_{j=1}^{\hat{R}} \in \arg\min_{(A_j, B_j)_{j=1}^{\hat{R}}} \frac{1}{N} \sum_{i=1}^N \| \sum_{j=1}^{\hat{R}} (A_j \otimes B_j) x_i - \hat{y}_i \|_1 \tag{7}$$

$$:= \arg\min_{(A_j, B_j)_j^{\hat{R}}} \mathcal{R}_\mathcal{L}(\sum_{j=1}^{\hat{R}} A_j \otimes B_j; \mathcal{D}), \tag{8}$$

where the dimensions of $A_i, B_i$ and rank $\hat{R}$ are hyper-parameters and needs to be fixed before solving equation 7. Our next theorem shows that $W^*$ might not be identifiable in the presence of noise under optimization problem equation 6.

**Theorem 4.1.** *If $N \le 0.1m$ and $0 < \eta < 0.5$, then there exists $\alpha = \mathcal{O}(t_0)$ such that with nonzero probability the following holds,*

$$\inf_{W:\|W-W^*\|_\infty \le \alpha} \mathcal{R}_\mathcal{L}(W; \mathcal{D}) < \mathcal{R}_\mathcal{L}(W^*; \mathcal{D}), \tag{9}$$

*Proof sketch.* Let $\mathcal{I} = \{i | \hat{y}_i \ne y_i, i = 1, 2, \dots, N\}$ and $\mathcal{I}^c = \{1, 2, \dots, N\} - \mathcal{I}$. Consider an $m$ by $n$ matrix $\Delta W$. Our goal is to find $\alpha$ and $\Delta W$ such that $\|\Delta W\|_\infty \le \alpha$ and $\mathcal{R}_\mathcal{L}(W^* + \Delta W) - \mathcal{R}_\mathcal{L}(W^*) \le 0$ with non-zero probability. We limit our analysis to $\Delta W$ with non-zero entries in the first row and zero entries in other rows. That is, $\Delta W_{i,j} = 0$ if $i > 1$. Therefore, we have

$$\mathcal{R}_\mathcal{L}(W^* + \Delta W) - \mathcal{R}_\mathcal{L}(W^*) = \tag{10}$$
$$\frac{1}{N} \sum_{i \in \mathcal{I}} \left( |\Delta W[1] \cdot x_i - \epsilon_i^{(1)}| - |\epsilon_i^{(1)}| \right) + \frac{1}{N} \sum_{i \in \mathcal{I}^c} |\Delta W[1] \cdot x_i|$$

where $\Delta W[1]$ is the first row of $\Delta W$ and $\epsilon_i^{(1)}$ is the first entry of noise vector $\epsilon_i$. By perturbing the first row of $W^*$, and by Theorem 1 of Ma & Fattahi (2022) for 1-layer linear model, there exists $\alpha = \mathcal{O}(t_0)$ and $\Delta W[1]$ such that, $\|\Delta W[1]\|_\infty \le \alpha$ and $\mathcal{R}_\mathcal{L}(W^* + \Delta W) - \mathcal{R}_\mathcal{L}(W^*) < 0$ $w.p.$ $\frac{1}{16}$. $\square$

Note that if Equation 9 holds, then the (sub-)gradient descent cannot find the true weight matrix $W^*$ with equation 6, and $W^*$ is not identifiable. This is because Equation 9 implies that $W^*$ is not a local minimum. This further implies that optimization problem equation 6 is vulnerable to noisy training dataset. This theorem also provides the following insights on why optimization problem equation 7 would be a better alternative compared to equation 6,

- Theorem 4.1 cannot be applied to a scenario where there are multiple ground truth solutions (the solution is not unique). In particular, if we change the optimization problem in a way that there would be multiple optimal solutions (e.g., optimization problem equation 7), then our proof technique and results cannot be used to show that all the possible solutions are not local minima. As a result, there is the possibility that some of the solutions in $\mathcal{S}$ remain local minima by optimization problem equation 7.

- We should avoid a convex objective function when $W^*$ is not identifiable. Since $\mathcal{R}_\mathcal{L}(W)$ is a convex objective function, the gradient descent always converges to a global optimal solution, and if $W^*$

is not a local minimum, the gradient descent never converges to $W^*$. On the other hand, if the loss landscape is non-convex, the gradient descent can converge to either global or local minima. Therefore, if some of the solutions in $\mathcal{S}$ are identifiable and are local minima, then there is a chance (depending on starting point) that gradient descent converges to those solutions under optimization problem equation 7.

Based on the above insight, we propose to improve robustness by replacing $W$ with $\sum_{i=1}^{\hat{R}} A_i \otimes B_i$ in equation 6. This creates multiple ground truth solutions (since $|\mathcal{S}| > 1$) and introduces non-convexity to the problem. We summarize our proposed solution in the next part.

## 4.2 Proposed Solution

In this section, we introduce a method based on Kronecker product decomposition to enhance the model robustness against noisy labels. In general, training a neural network can be formulated as follows,

$$\hat{W}^{[1]}, \dots, \hat{W}^{[L]} = \arg \min_{W^{[1]}, \dots, W^{[L]}} \mathcal{R}_{\mathcal{L}}(W^{[1]}, \dots, W^{[L]}; \mathcal{D}). \tag{11}$$

However, based on our insight from Theorems 4.1, we replace $W^{[l]}$ by $\sum_{i=1}^{r_l} A_i^{[l]} \otimes B_i^{[l]}$ in equation 11 and solve the following optimization problem,

$$\min_{[A_i^{[l]}, B_i^{[l]}]_{i \leq r_l, l \leq L}} \mathcal{R}_{\mathcal{L}}(\sum_{i=1}^{r_1} A_i^{[1]} \otimes B_i^{[1]}, \dots, \sum_{i=1}^{r_L} A_i^{[L]} \otimes B_i^{[L]}; \mathcal{D}), \tag{12}$$

It is worth noting that for layers where the weight is a tensor (e.g., convolutional layers), we first convert the tensor into a matrix and then write a KPD. For example, a convolutional kernel with dimensions $(3, 3, 256, 512)$ is reshaped into a matrix of size $(3 \times 256, 3 \times 512)$, and then Kronecker product decomposition is applied for factorization.

---

**Algorithm 1** Training with KPD

1: **Inputs**: Training dataset $\mathcal{D}$, Number of layers $L$, $r_1, r_2, \dots, r_L$, Learning rate $\gamma$, Number of epochs $T$
2: Initialize $A_j^{[l]}, B_j^{[l]}, 1 \leq j \leq r_l$ randomly $\forall 1 \leq l \leq L$
3: **for** epoch $t = 1$ to $T$ **do**
4:     Shuffle $\mathcal{D}$ to randomize mini-batches
5:     **for** each mini-batch $\mathcal{B} \subset \mathcal{D}$ **do**
6:         Compute batch loss $\mathcal{R}_{\mathcal{L}}(\sum_{i=1}^{r_1} A_i^{[1]} \otimes B_i^{[1]}, \dots, \sum_{i=1}^{r_L} A_i^{[L]} \otimes B_i^{[L]}; \mathcal{B})$
7:         Compute gradients with respect to all $A_j^{[l]}, B_j^{[l]}, 1 \leq j \leq r_l, 1 \leq l \leq L$
8:         Update weights using gradient descent:

$$A_j^{[l]} \leftarrow A_j^{[l]} - \gamma \nabla_{A_j^{[l]}} \mathcal{R}_{\mathcal{L}}, \quad B_j^{[l]} \leftarrow B_j^{[l]} - \gamma \nabla_{B_j^{[l]}} \mathcal{R}_{\mathcal{L}}$$

9:     **end for**
10: **end for**
11: $W^{[l]} \leftarrow \sum_{j=1}^{r_l} A_j^{[l]} \otimes B_j^{[l]}$ for all $1 \leq l \leq L$
12: **return** $W^{[l]}, 1 \leq l \leq L$

---

To solve problem equation 12, we follow Algorithm 1. Note that, in Algorithm 1, we use a robust loss function in addition to leveraging Kronecker product decomposition. The advantage of Algorithm 1 is that it can be combined with any robust loss function available in the literature. We will study the impact of KPD on different robust loss functions in the experiment section.

# 5 Experiments

In this section, we will discuss our experiments for both regression and classification tasks on different datasets. As we stated in Algorithm 1, our method can be applied to arbitrary loss functions, including the robust loss functions. Therefore, we will conduct experiments with different combinations.

## 5.1 Experiment on Linear Regression

**Synthetic Data:** We conduct an experiment with linear regression model to confirm KPD can improve robustness. We generate 200 data samples, including 100 for training, 100 for testing. The input and output dimensions are 36 (i.e., $x_i \in \mathbb{R}^{36}, y_i \in \mathbb{R}^{36}$). Output $y_i$ is generated by $y_i = W^* x_i$, where $W^*$ is the ground truth matrix generated randomly before generating the data points. For the training dataset, we set a noise ratio $\eta_i$ as 0.55 for any $i$, which means 55% of the training data points would be impacted by a Gaussian noise. The covariance matrix of the Gaussian noise is set to be $60I$, where $I$ is a 36 by 36 identity matrix. Note that the test dataset does not include any noise as we want to know whether the model trained under noisy dataset can generalize to unseen clean data or not. For the normal model without KPD, we solve the following optimization problem, $\min_W \frac{1}{100} \sum_{i=1}^{100} \|\hat{y}_{i(train)} - W x_{i(train)}\|_1$. For the model using KPD, we replace $W$ with 25 matrices $A_j \in \mathbb{R}^{6 \times 6}$ and $B_j \in \mathbb{R}^{6 \times 6}$. We get the optimal model by solving the following problem, $\min_{A_j, B_j, j \leq 25} (\frac{1}{100} \sum_{i=1}^{100} \|\hat{y}_{i(train)} - \sum_{j=1}^{25} (A_j \otimes B_j) x_{i(train)}\|_1)$. Figure 1 shows the comparison of the normal linear model (no decomposition) and the model with the KPD using the MAE loss. As we expected, while the training errors are the same under KPD and normal model, the model with KPD generalizes better to unseen clean data (i.e., lower test loss).

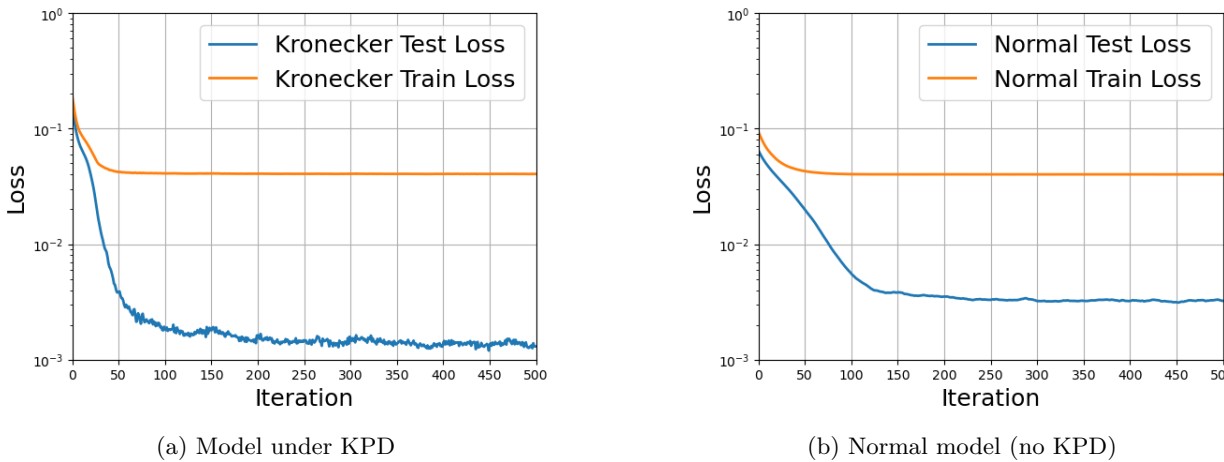

(a) Model under KPD                    (b) Normal model (no KPD)

Figure 1: Comparison of the loss with and without KPD.

**California Housing Experiments:** We conduct an experiment with California Housing dataset (Pace & Barry, 1997) for the regression task. We split the dataset randomly into train (10000 samples), validation (3000 samples), and test (3000 samples) dataset for the experiment. We add zero mean Gaussian noise with standard deviation 3 to $y_i$ in train and validation dataset. We use the validation dataset to find the optimal number of epochs for training and avoid over fitting. We repeat the experiment 5 times and report mean and standard deviation of three different kind of robust loss over the 5 runs. For the Kronecker product decomposition, we use a rank of 100, and $A_i$ and $B_i$ are 2 by 1 and 4 by 1, respectively. The results in terms of Mean Absolute Error (MAE) are shown in Table 1. As we can see in this table, under all of the robust loss functions, we can get enhancements in performance with KPD, especially with Huber Loss and Tukey Loss.

Table 1: Results for California housing regression

| Loss function | Normal model | KPD model |
|---|---|---|
| $l_1$ Loss | $4.326 \pm 0.541$ | $4.013 \pm 0.390$ |
| Huber Loss | $4.329 \pm 0.800$ | $3.137 \pm 0.711$ |
| Tukey Loss | $4.125 \pm 0.732$ | $3.097 \pm 0.861$ |

## 5.2 Classification result of Experiment on synthetic dataset

In this section, we verify the effectiveness of our algorithm for classification tasks. We use three image classification datasets. The detailed experimental settings for each dataset are as follows.

Table 2: Performance of different loss functions under normal, KPD, and Rank Factorization models with different noise rates for various datasets.

| Datasets | Loss Functions | Normal Model | | KPD | | Rank Factorization | |
|---|---|---|---|---|---|---|---|
| | | 0.4 | 0.6 | 0.4 | 0.6 | 0.4 | 0.6 |
| **Symmetric Noise** | | | | | | | |
| FASHION MNIST | Cross Entropy (CE) | $89.65 \pm 0.11$ | $\underline{88.10 \pm 0.59}$ | $\mathbf{91.65 \pm 0.03}$ | $87.28 \pm 0.05$ | $86.04 \pm 0.05$ | $86.48 \pm 0.08$ |
| | Symmetric CEWang et al. (2019) | $92.38 \pm 0.04$ | $90.15 \pm 0.08$ | $\mathbf{92.49 \pm 0.08}$ | $\underline{90.88 \pm 0.09}$ | $92.07 \pm 0.05$ | $90.48 \pm 0.02$ |
| | Normalized CEMa et al. (2020) | $89.75 \pm 0.02$ | $87.91 \pm 0.04$ | $\mathbf{91.21 \pm 0.04}$ | $\underline{89.54 \pm 0.06}$ | $88.10 \pm 0.04$ | $84.99 \pm 0.02$ |
| | Trunc $L_q$Zhang & Sabuncu (2018) | $90.91 \pm 0.04$ | $88.77 \pm 0.04$ | $\mathbf{91.55 \pm 0.06}$ | $\underline{88.78 \pm 0.01}$ | $90.42 \pm 0.05$ | $87.17 \pm 0.03$ |
| | $L_{dmi}$ Xu et al. (2019) | $89.47 \pm 0.09$ | $87.02 \pm 0.06$ | $\mathbf{90.25 \pm 0.06}$ | $\underline{89.96 \pm 0.19}$ | $88.10 \pm 0.06$ | $84.99 \pm 0.14$ |
| | Asymmetric GCE Zhou et al. (2021) | $90.06 \pm 0.08$ | $88.60 \pm 0.13$ | $\mathbf{91.34 \pm 0.24}$ | $\underline{89.76 \pm 0.27}$ | $90.12 \pm 0.85$ | $86.73 \pm 2.24$ |
| | LDR KL Zhu et al. (2023) | $91.14 \pm 0.14$ | $87.79 \pm 1.75$ | $\mathbf{92.97 \pm 0.39}$ | $88.85 \pm 1.40$ | $91.34 \pm 1.31$ | $\underline{88.94 \pm 0.30}$ |
| | $\epsilon$ -softmaxJialiang et al. (2024) | $90.57 \pm 0.09$ | $88.91 \pm 1.02$ | $\mathbf{92.37 \pm 0.88}$ | $\underline{89.47 \pm 0.46}$ | $89.63 \pm 0.57$ | $86.43 \pm 0.50$ |
| CIFAR-10 | Cross Entropy (CE) | $75.44 \pm 6.40$ | $64.04 \pm 5.67$ | $\mathbf{75.52 \pm 5.16}$ | $63.55 \pm 3.88$ | $73.30 \pm 0.25$ | $\underline{65.64 \pm 0.05}$ |
| | Symmetric CE | $88.58 \pm 1.40$ | $81.29 \pm 1.77$ | $\mathbf{88.95 \pm 1.03}$ | $\underline{82.58 \pm 1.50}$ | $84.13 \pm 0.05$ | $75.51 \pm 0.02$ |
| | Normalized CE | $77.98 \pm 0.12$ | $69.31 \pm 0.25$ | $\mathbf{82.42 \pm 0.05}$ | $\underline{72.20 \pm 0.25}$ | $76.60 \pm 0.07$ | $69.13 \pm 0.02$ |
| | Trunc $L_q$ | $80.28 \pm 0.04$ | $66.62 \pm 0.07$ | $\mathbf{82.52 \pm 0.04}$ | $\underline{69.57 \pm 0.04}$ | $71.87 \pm 0.09$ | $57.70 \pm 0.06$ |
| | $L_{dmi}$ | $80.00 \pm 0.02$ | $67.11 \pm 0.06$ | $\mathbf{81.44 \pm 0.01}$ | $\underline{70.19 \pm 0.02}$ | $72.51 \pm 0.01$ | $62.44 \pm 0.01$ |
| | Asymmetric GCE | $79.48 \pm 1.38$ | $68.97 \pm 0.96$ | $\mathbf{81.69 \pm 0.06}$ | $\underline{72.83 \pm 0.46}$ | $77.51 \pm 0.27$ | $61.33 \pm 0.46$ |
| | LDR KL | $82.01 \pm 0.13$ | $70.72 \pm 0.36$ | $\mathbf{84.53 \pm 0.16}$ | $\underline{75.37 \pm 0.09}$ | $79.27 \pm 0.53$ | $64.46 \pm 0.10$ |
| | $\epsilon$ -softmax | $82.26 \pm 0.26$ | $65.88 \pm 0.25$ | $\mathbf{85.08 \pm 0.08}$ | $\underline{70.93 \pm 1.38}$ | $78.77 \pm 0.93$ | $62.47 \pm 0.71$ |
| CIFAR-100 | Cross Entropy (CE) | $32.65 \pm 0.95$ | $23.80 \pm 0.10$ | $33.70 \pm 0.05$ | $\underline{27.29 \pm 0.08}$ | $\mathbf{35.13 \pm 0.09}$ | $24.72 \pm 0.06$ |
| | Symmetric CE | $39.57 \pm 0.06$ | $27.35 \pm 0.04$ | $42.22 \pm 0.01$ | $\underline{27.98 \pm 0.06}$ | $\mathbf{42.80 \pm 0.03}$ | $27.84 \pm 0.05$ |
| | Normalized CE | $39.08 \pm 0.05$ | $26.37 \pm 0.05$ | $41.14 \pm 0.04$ | $27.12 \pm 0.02$ | $\mathbf{42.22 \pm 0.09}$ | $\underline{28.91 \pm 0.03}$ |
| | Trunc $L_q$ | $39.50 \pm 0.04$ | $27.39 \pm 0.02$ | $42.28 \pm 0.08$ | $\underline{27.94 \pm 0.02}$ | $\mathbf{42.81 \pm 0.06}$ | $27.86 \pm 0.08$ |
| | $L_{dmi}$ | $39.64 \pm 0.04$ | $27.26 \pm 0.08$ | $42.28 \pm 0.06$ | $\underline{27.89 \pm 0.09}$ | $\mathbf{42.71 \pm 0.10}$ | $27.36 \pm 0.07$ |
| | Asymmetric GCE | $34.24 \pm 0.13$ | $23.19 \pm 0.04$ | $\mathbf{44.09 \pm 0.22}$ | $\underline{28.37 \pm 0.16}$ | $39.87 \pm 0.11$ | $25.64 \pm 0.06$ |
| | LDR KL | $33.27 \pm 0.32$ | $23.47 \pm 0.34$ | $33.69 \pm 0.02$ | $20.84 \pm 0.22$ | $\mathbf{35.97 \pm 0.10}$ | $\underline{24.63 \pm 0.17}$ |
| | $\epsilon$ -softmax | $37.65 \pm 0.04$ | $24.69 \pm 0.06$ | $\mathbf{50.49 \pm 0.02}$ | $\underline{36.31 \pm 0.08}$ | $32.67 \pm 0.17$ | $21.79 \pm 0.19$ |
| **Asymmetric Noise** | | | | | | | |
| FASHION MNIST | Cross Entropy (CE) | $88.72 \pm 0.80$ | $58.21 \pm 0.51$ | $88.36 \pm 0.88$ | $\underline{59.17 \pm 0.86}$ | $\mathbf{89.25 \pm 0.57}$ | $57.86 \pm 0.18$ |
| | Symmetric CEWang et al. (2019) | $89.23 \pm 0.74$ | $58.29 \pm 0.82$ | $88.72 \pm 0.72$ | $\underline{59.12 \pm 0.59}$ | $\mathbf{89.60 \pm 1.04}$ | $57.50 \pm 0.13$ |
| | Normalized CEMa et al. (2020) | $75.07 \pm 0.16$ | $56.75 \pm 0.05$ | $\mathbf{77.26 \pm 2.52}$ | $\underline{56.91 \pm 0.10}$ | $74.02 \pm 0.11$ | $56.37 \pm 0.09$ |
| | Trunc $L_q$Zhang & Sabuncu (2018) | $88.81 \pm 0.46$ | $58.06 \pm 0.55$ | $\mathbf{89.34 \pm 0.64}$ | $\underline{58.26 \pm 0.08}$ | $88.19 \pm 0.46$ | $57.27 \pm 0.11$ |
| | $L_{dmi}$ Xu et al. (2019) | $88.98 \pm 0.73$ | $59.24 \pm 0.82$ | $\mathbf{89.13 \pm 0.69}$ | $\underline{59.26 \pm 0.13}$ | $89.03 \pm 0.77$ | $57.68 \pm 0.35$ |
| | Asymmetric GCE | $88.87 \pm 0.13$ | $59.56 \pm 0.37$ | $89.21 \pm 0.16$ | $59.77 \pm 0.08$ | $\mathbf{89.36 \pm 0.05}$ | $58.33 \pm 0.16$ |
| | LDR KL | $89.19 \pm 0.28$ | $57.47 \pm 0.18$ | $\mathbf{90.88 \pm 0.03}$ | $58.26 \pm 0.08$ | $88.87 \pm 0.06$ | $57.85 \pm 0.26$ |
| | $\epsilon$ -softmax | $89.13 \pm 0.12$ | $56.31 \pm 0.28$ | $\mathbf{90.51 \pm 0.04}$ | $57.21 \pm 0.19$ | $89.59 \pm 0.16$ | $57.37 \pm 0.04$ |
| CIFAR-10 | Cross Entropy (CE) | $74.18 \pm 1.06$ | $59.09 \pm 1.86$ | $\mathbf{75.97 \pm 1.29}$ | $\underline{59.12 \pm 0.86}$ | $75.38 \pm 0.91$ | $56.19 \pm 1.18$ |
| | Symmetric CE | $83.33 \pm 6.18$ | $71.55 \pm 11.68$ | $85.00 \pm 5.10$ | $71.76 \pm 12.10$ | $\mathbf{85.70 \pm 0.01}$ | $\underline{79.62 \pm 0.03}$ |
| | Normalized CE | $71.94 \pm 0.05$ | $50.43 \pm 0.08$ | $\mathbf{74.86 \pm 0.01}$ | $\underline{51.65 \pm 0.08}$ | $66.94 \pm 0.10$ | $48.42 \pm 0.07$ |
| | Trunc $L_q$ | $80.28 \pm 0.04$ | $66.62 \pm 0.07$ | $\mathbf{82.52 \pm 0.04}$ | $\underline{69.57 \pm 0.04}$ | $72.32 \pm 0.07$ | $51.86 \pm 0.04$ |
| | $L_{dmi}$ | $82.42 \pm 6.18$ | $72.51 \pm 11.68$ | $\mathbf{85.78 \pm 5.10}$ | $72.16 \pm 12.10$ | $85.23 \pm 0.09$ | $\underline{79.87 \pm 0.04}$ |
| | Asymmetric GCE | $75.89 \pm 0.05$ | $65.27 \pm 0.16$ | $\mathbf{76.17 \pm 0.10}$ | $66.28 \pm 0.09$ | $76.16 \pm 0.07$ | $\underline{66.37 \pm 0.06}$ |
| | LDR KL | $76.21 \pm 0.85$ | $\underline{56.27 \pm 2.02}$ | $\mathbf{82.30 \pm 1.42}$ | $54.97 \pm 0.37$ | $79.41 \pm 0.03$ | $52.47 \pm 0.25$ |
| | $\epsilon$ -softmax | $73.76 \pm 0.12$ | $61.77 \pm 2.53$ | $\mathbf{80.11 \pm 3.50}$ | $\underline{63.08 \pm 1.11}$ | $74.45 \pm 0.20$ | $57.27 \pm 0.31$ |
| CIFAR-100 | Cross Entropy (CE) | $34.11 \pm 0.07$ | $23.21 \pm 0.07$ | $\mathbf{38.06 \pm 0.02}$ | $\underline{25.71 \pm 1.57}$ | $37.98 \pm 0.06$ | $21.79 \pm 0.06$ |
| | Symmetric CE | $39.39 \pm 0.03$ | $28.17 \pm 0.09$ | $41.42 \pm 0.10$ | $\underline{30.42 \pm 0.25}$ | $\mathbf{42.20 \pm 0.09}$ | $27.62 \pm 0.04$ |
| | Normalized CE | $38.78 \pm 0.10$ | $\underline{28.38 \pm 0.06}$ | $41.29 \pm 0.04$ | $24.26 \pm 0.47$ | $\mathbf{42.17 \pm 0.06}$ | $27.16 \pm 0.07$ |
| | Trunc $L_q$ | $39.74 \pm 0.09$ | $26.94 \pm 0.08$ | $42.20 \pm 0.06$ | $27.26 \pm 0.03$ | $\mathbf{43.25 \pm 0.09}$ | $\underline{28.27 \pm 0.04}$ |
| | $L_{dmi}$ | $\mathbf{35.14 \pm 0.00}$ | $17.70 \pm 0.00$ | $34.09 \pm 0.00$ | $15.49 \pm 0.00$ | $34.23 \pm 0.01$ | $\underline{18.62 \pm 0.02}$ |
| | Asymmetric GCE | $36.78 \pm 0.11$ | $27.69 \pm 0.04$ | $\mathbf{40.35 \pm 0.09}$ | $\underline{28.47 \pm 0.08}$ | $35.31 \pm 0.09$ | $27.98 \pm 0.15$ |
| | LDR KL | $33.13 \pm 0.58$ | $20.56 \pm 0.08$ | $\mathbf{40.91 \pm 0.10}$ | $\underline{24.93 \pm 0.37}$ | $34.27 \pm 1.12$ | $20.59 \pm 1.16$ |
| | $\epsilon$ -softmax | $37.71 \pm 0.47$ | $20.92 \pm 0.30$ | $\mathbf{43.42 \pm 2.23}$ | $\underline{25.14 \pm 0.16}$ | $34.21 \pm 0.12$ | $19.14 \pm 1.85$ |

**FashionMNIST:** The original FashionMNIST dataset (Xiao et al., 2017) has 60,000 training samples and 10,000 testing samples. We use 10,000 samples of training samples to create a validation set and the remaining 50,000 samples as a training set. We conduct experiments under noise ratio $\eta$ equal to 0.4 and 0.6 (Note that the noise will be added to training and validation dataset). We consider two types of noise, symmetric noise and asymmetric noise. For symmetric noise, an image with label $y$ is assigned to label $y' \neq y$, with probability $\eta$. For asymmetric noise, we pre-define a set of mappings $\{y \mapsto y'\}$ based on Patrini et al. (2017). In this case, label $y$ will be corrupted and mapped to new label $y'$ based on the pre-defined mapping with probability $\eta$. We adopt ResNet-18 (He et al., 2016) as the backbone for the classifier. RandomCrop and RandomHorizontalFlip (Chen et al., 2020) are used for data augmentation when training the classifier.

**CIFAR10:** CIFAR-10 (Krizhevsky et al., 2009) contains 50,000 training samples and 10,000 test samples from 10 classes. We split the training samples randomly and create a train set with 40,000 samples and a validation set with 10,000. The noisy dataset is created similar to FashionMNIST, except for the mapping set, which still follows Patrini et al. (2017) for asymmetric noise. Instead of ResNet-18, we use ResNet-50 (He et al., 2016) for image classification. RandomCrop, RandomHorizontalFlip and Cutout (DeVries, 2017) are used for data augmentation for training.

**CIFAR100:** CIFAR-100 (Krizhevsky et al., 2009) contains 50,000 training samples and 10,000 test samples from 100 classes. Similar to CIFAR10 and FasionMNIST, we create a validation set by picking 10,000 sample randomly from the training samples. The symmetric noise is generated similar to the noise generated for CFAR10. For the asymmetric noise, we divide the 100 classes into 20 super-classes. The $j$-th super-class has five classes $\{y^{j(1)}, ..., y^{j(5)}\}$. Label $y^{j(i)}$ will be kept unchanged with probability $1 - \eta$ and change to $y^{j(i+1)}$ with probability $\eta$.[1] The other experiments settings are the same as CIFAR-10 dataset.

In this part, we initially set the learning rate as 0.005. We use a batch size of 512 and Adam optimizer (Kingma & Ba, 2017). We use a cosine-scheduled learning rate that starts at 0.005 and gradually decreases to 0.001. The best number of epochs found based on the accuracy on the validation set.

We train the model under three different scenario. First, we use optimization problem equation 11 to train the model. We report the accuracy for optimization problem equation 11 under *Normal Model* column in Table 2. Second, we use optimization problem equation 12 with $r_l = 100 \ \forall \ l \leq L$ to train the model and report the results under *KPD* column in Table 2. We set the size of $A_i^{[l]}$ and $B_i^{[l]}$ to integers that are nearest to the square root of the dimension of $W^{[l]}$. As we mentioned in Section 3.2, rank factorization is a special case of KPD. As result, for completeness, as the third scenario, we repeat the experiment with rank factorization. In particular, we replace $W^{[l]}$ by $U^{[l]} \cdot V^{[l]}$ in equation 11 and solve the following optimization problem, $\min_{[U^{[l]}, V^{[l]}]_{l \leq L}} \mathcal{R}_{\mathcal{L}}(U^{[1]} \cdot V^{[1]}, \ldots, U^{[L]} \cdot V^{[L]}, \mathcal{D})$, where, $W^{[l]}$ is $m^{[l]}$ by $n^{[l]}$, and $U^{[l]}$ is $m^{[l]}$ by $r$ and $V^{[l]}$ is $r$ by $n^{[l]}$. In our experiment, $r = 100$, and $m^{[l]}$ and $n^{[l]}$ are determined by the ResNet architecture. The results for rank factorization are reported under *Rank Factorization* column in Table 2. Note that, in all these scenarios, we evaluate the performance of the model by the accuracy on the test clean dataset. We repeat the experiment for 3 times to report mean and standard deviation of the accuracy. Moreover, we also repeat the experiment for different loss function $\mathcal{L}$ and report the results in different rows.

As shown in Table 2, under the same loss function, the model based on KPD can achieve a similar or better accuracy compared to the normal model. For example, for the model trained on CIFAR-10 dataset with Normalized CE loss under 0.4 noise ratio and symmetric noise, KPD can increase the accuracy by 4.44 percent points and for the model trained on CIFAR-100 dataset with $\epsilon$-softmax under 0.4 noise ratio and symmetric noise, KPD can increase the accuracy by 12.48 percent points. Moreover, when the noise level is 0.4, our method (rank factorization or KPD) can improve performance for 47 out of 48 rows. When noise ratio is 0.6, our algorithm increases the accuracy compared to the normal model in 45 rows out of 48 rows. This observation verifies that using our algorithm is more likely to find a better optimal point when the noise level is smaller.

---

[1] $y^{j(5)}$ will change to $y^{j(1)}$

## 5.3 Experiment on Real-world noisy dataset

**Clothing1M:** Clothing1M (Xiao et al., 2015) is a large-scale dataset designed for learning with real-world noisy labels. It contains 1 million images of clothing items across 14 categories (e.g., T-shirt, Shirt, Knitwear) collected from online shopping websites. The labels are generated from the surrounding text metadata, resulting in an estimated noise rate of approximately 38.5 %. Unlike synthetic noise in CIFAR-100, the noise in Clothing1M is instance-dependent and reflects real-world label corruption. In addition to the 1M noisy training samples, the dataset provides 50k clean training samples, 14k clean validation samples, and 10k clean test samples for model evaluation. In our experiments, we primarily use the 1M noisy samples for training and evaluate the performance on the 10k clean test set, following the standard experimental protocol.

We train the model under two different scenarios, we also report the accuracy for optimization problem equation 11 under *Normal Model* and compare it with the optimization problem equation 12 under *KPD* decomposed model. The experiment results show that our method can still outperform with cross entropy loss function and $\epsilon$-softmax loss function. Since the label noise in Clothing1M is not synthetic but collected from real-world metadata, this experiment demonstrates that the robustness of our method generalizes beyond controlled noise settings and remains effective in practical scenarios.

Table 3: Performance comparison of different loss functions on the Clothing1M dataset (Real-world Noise) under Normal and KPD.

| Dataset | Loss Functions | Normal Model | KPD |
|---|---|---|---|
| | | Test Accuracy (%) | Test Accuracy (%) |
| Clothing1M | Cross Entropy (CE) | $68.91 \pm 0.21$ | $69.16 \pm 0.52$ |
| | $\epsilon$-softmaxJialiang et al. (2024) | $69.16 \pm 0.37$ | $70.54 \pm 1.26$ |

## 5.4 Ablation study

The robustness of a model can be influenced by its parameter size (Rolnick et al., 2018). To investigate the impact of model parameter size (number of learning paramters) on robustness, we design experiments focusing on two hyper-parameters of KPD: *rank* and *patch size*. In Eq 3, $R$ denotes the rank and the shape of $B_i$ denotes the patch size. Both of them can affect the model parameter size.All the ablation studies are conducted under the same setting as the classification experiments with cross entropy loss on CIFAR-10 and symmetric noise.

Table 4: Results for ablation study in rank.
Patch size is fixed and equal to (4,4).

| Noise Ratio | Rank ($r_l, l \leq L$) | Accuracy |
|---|---|---|
| 0.4 | 20 | $58.68 \pm 4.34$ |
| 0.4 | 50 | $70.93 \pm 3.29$ |
| 0.4 | 100 | $75.44 \pm 6.40$ |
| 0.4 | 150 | $74.66 \pm 0.49$ |
| 0.4 | 200 | $74.78 \pm 0.51$ |

**Rank:** As shown in Table 4, the accuracy of the model increases as the rank increases. This can be explained by Rolnick et al. (2018) that larger network architecture tends to be more robust to label noise. This experiment also implies that the rank significantly play important role in success of KPD for robustness. In other words, KPD can improve robustness if we use large rank during the training. This observation is consistent with our theoretical analysis in Section 4.1. According to Theorem 4.1, in the presence of label noise, the ground truth weight matrix $W^*$ is not necessarily the unique or global minimizer of the noisy empirical risk. As the rank $R$ increases, the hypothesis space of the Kronecker Product Decomposition

(KPD) expands, providing the model with excessive degrees of freedom. This increased capacity allows the optimization process to be more easily "trapped" by non-ground truth solutions that fit the label noise rather than the underlying clean distribution.

In contrast, a lower rank $R$ acts as a form of structural regularization (or implicit bias). By constraining the search space, KPD effectively filters out complex noise patterns that cannot be efficiently represented in a low-rank Kronecker form, thereby maintaining better generalization. This suggests that the choice of $R$ involves a trade-off between model capacity and noise robustness.

Table 5: Results for ablation study in patch size.
Rank is fixed and equal to 100.

| Noise Ratio | Patch Size | Accuracy |
|---|---|---|
| 0.4 | (2,2) | $79.80 \pm 3.45$ |
| 0.4 | (4,4) | $75.44 \pm 6.40$ |
| 0.4 | (8,8) | $74.23 \pm 1.36$ |

**Patch Size:** We adopt three different patch sizes for all the layers: 2 by 2, 4 by 4, and 8 by 8. As shown in the Table5, as the patch size increases, the performance drops. This indicates that a model with more parameters is more robust to the noisy label.

## 6  Conclusion

In this paper, we propose a novel method for enhancing the robustness of deep learning models by leveraging Kronecker Product Decomposition (KPD) or its special case, Rank Factorization. Our approach is flexible, as it can be used as a plug-in with other robust techniques, such as robust loss functions. Through extensive experiments on both synthetic and real-world datasets, we confirm that Kronecker decomposition enhances model robustness across different datasets, noise levels, and training conditions. Additionally, our ablation study reveals that the model's performance is influenced by factors such as decomposition rank and patch size.

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

# A  Proof

*Theorem 4.1.* Let $\mathcal{I} = \{i|\hat{y}_i \neq y_i, i = 1, 2, \ldots, N\}$ and $\mathcal{I}^c = \{1, 2, \ldots, N\} - \mathcal{I}$. Consider an $m$ by $n$ matrix $\Delta W$. Our goal is to find $\alpha$ and $\Delta W$ such that $||\Delta W||_\infty \leq \alpha$ and $\mathcal{R}_\mathcal{L}(W^* + \Delta W) - \mathcal{R}_\mathcal{L}(W^*) \leq 0$ with non-zero probability. We have,

$$\mathcal{R}_\mathcal{L}(W^*) = \frac{1}{N} \sum_{i \in \mathcal{I}} ||\epsilon_i||_1, \quad \mathcal{R}_\mathcal{L}(W^* + \Delta W) = \frac{1}{N} \sum_{i \in \mathcal{I}} ||\Delta W \cdot x_i - \epsilon_i||_1 + \frac{1}{N} \sum_{i \in \mathcal{I}^c} ||\Delta W \cdot x_i||_1 \implies$$

$$\mathcal{R}_{\mathcal{L}}(W^* + \Delta W) - \mathcal{L}(W^*) = \frac{1}{N} \sum_{i \in \mathcal{I}} (||\Delta W \cdot x_i - \epsilon_i||_1 - ||\epsilon_i||_1) + \frac{1}{N} \sum_{i \in \mathcal{I}^c} ||\Delta W \cdot x_i||_1 \quad (13)$$

We limit our analysis to $\Delta W$'s with non-zero entries in the first row and zero entries in other rows. That is, $\Delta W_{i,j} = 0$ if $i > 1$. Therefore, equation 13, can be written as follows,

$$\mathcal{R}_{\mathcal{L}}(W^* + \Delta W) - \mathcal{R}_{\mathcal{L}}(W^*) = \frac{1}{N} \sum_{i \in \mathcal{I}} (|\Delta W[1] \cdot x_i - \epsilon_i^{(1)}| - |\epsilon_i^{(1)}|) + \frac{1}{N} \sum_{i \in \mathcal{I}^c} |\Delta W[1] \cdot x_i|, \quad (14)$$

where $\Delta W[1]$ is the first row of $\Delta W$ and $\epsilon_i^{(1)}$ is the first entry of noise vector $\epsilon_i$. By perturbing only the first row of $W^*$, and by Theorem 1 of Ma & Fattahi (2022) for 1-layer linear model, when $N \leq 0.1m$, there exists $\alpha = \mathcal{O}(t_0)$ and $\Delta W[1]$ such that $||\Delta W[1]||_\infty \leq \alpha$,

$$\mathcal{R}_{\mathcal{L}}(W^* + \Delta W) - \mathcal{R}_{\mathcal{L}}(W^*) = \frac{1}{N} \sum_{i \in \mathcal{I}} (|\Delta W[1] \cdot x_i - \epsilon_i^{(1)}| - |\epsilon_i^{(1)}|) + \frac{1}{N} \sum_{i \in \mathcal{I}^c} |\Delta W[1] \cdot x_i| \leq -c \cdot (p_0 \eta \alpha)$$

$$w.p. \quad \frac{1}{16},$$

where $c$ is a positive constant. The above equation implies that $W^* + \Delta W$ can achieves a lower empirical loss compared to $W^*$ with probability at least $\frac{1}{16}$. As a result, $W^*$ is not identifiable with non-zero probability.

$\square$

## B  Loss functions

**MAE Loss (Regression)**: Mean Absolute Error (MAE) can improve robustness compared to mean squared error. It is defined as follows, $\mathcal{L}_{mae}(\tilde{y}, y) = ||\tilde{y} - y||_1$, where $\tilde{y}$ is the prediction of the ML model.

**Huber Loss (Regression) (Huber, 1992b)**: Huber loss combines the best properties of the mean absolute error and mean squared error by being quadratic for small errors and linear for large errors. Specifically, the formula of Huber loss is as follows:

$$\mathcal{L}_{huber}^{\delta}(\tilde{y}, y) = \begin{cases} \frac{1}{2}(\tilde{y} - y)^2, & \text{if } |\tilde{y} - y| \leq \delta, \\ \delta \cdot (|\tilde{y} - y| - \frac{1}{2}\delta), & \text{if } |\tilde{y} - y| > \delta. \end{cases} \quad (15)$$

where $\delta$ is a pre-defined constant. When $y$ is a vector, it is common to use the sum of Huber loss over all entries of $y$ (He et al., 2023).

**Tukey Loss (Regression) (Belagiannis et al., 2015)**: Tukey loss is another loss function for regression tasks. It is robust to the outliers since it is a constant if error $|\tilde{y} - y|$ larger than a certain threshold. The Tukey loss is defined as follows,

$$\mathcal{L}_{Tukey}^{\delta}(\tilde{y}, y) = \begin{cases} \frac{\delta^2}{6} \cdot (1 - (1 - \frac{(\tilde{y} - y)^2}{\delta^2})^3), & \text{if } |\tilde{y} - y| \leq \delta, \\ \frac{\delta^2}{6}, & \text{if } |\tilde{y} - y| > \delta, \end{cases} \quad (16)$$

where $\delta$ is a pre-defined constant. Again, an element-wise Tukey Loss can be used when $y$ is a vector (Li et al., 2021).

**Symmetric Cross Entropy (Classification) (Wang et al., 2019)**: To introduce symmetric cross entropy loss, first we need to introduce Reverse Cross Entropy as follows,

$$\mathcal{L}_{rce} = - \sum_{k=0}^{K-1} p(k|x) \log q(k|x) \quad (17)$$

where $q(k|x)$ is the ground truth probability that $x$ belongs to class $k$ and $p(k|x)$ is the predictive probability that $x$ belongs to class $k$. Then, the symmetric cross entropy (SCE) is the combination of the Reserve Cross-Entropy and the cross entropy formulated as follows,

$$\mathcal{L}_{sce} = \alpha_{sce}\mathcal{L}_{rce} + \beta_{sce}\mathcal{L}_{ce}, \tag{18}$$

where $\beta_{sce}$ is a hyper-parameter.

**Generalized Cross Entropy Loss (Classification) (Zhang & Sabuncu, 2018)**: The generalized cross entropy loss is a truncated version of $\mathcal{L}_q$ function given by

$$\mathcal{L}_{trunc}(f_k(x), k)) = \begin{cases} \mathcal{L}_q(\tau) & \text{if} f_k(x) \leq \tau \\ \mathcal{L}_q(f_k(\tau), k) & \text{if} f_k(x) > \tau \end{cases} \tag{19}$$

where $\tau$ is the threshold of the truncated loss function, $f_k(x)$ is predictive likelihood for class $k$. $\mathcal{L}_q$ function is given by

$$\mathcal{L}_q(f_k(x), k) = \frac{(1 - f_k(x)^q)}{q} \tag{20}$$

where $q$ is a hyper-parameter.

$\mathcal{L}_{dmi}$ **Loss (Classification) (Xu et al., 2019)**: $\mathcal{L}_{dmi}$ Loss is proposed based on the Shannon mutual information as the performance measure for classifiers. This loss function ensures that the optimal parameters always achieve the lowest loss in the presence of label noise. The formula can be written as:

$$\mathcal{L}_{\text{DMI}}(f(x), y) = -\log(|\det(Q(f(X), \mathcal{Y})|) \tag{21}$$

where $Q(f(X), \mathcal{Y})$ is the matrix format of the joint distribution of the neural network output and ground truth label. And det denotes the determinant of a matrix.

**Normalized Cross Entropy (Classification) (Ma et al., 2020)**: The normalized cross entropy is a loss function which satisfies the equation $\sum_k \mathcal{L}(f(x), k) = C, \forall f, x$, where $C$ is a constant. To satisfy this equation, the normalized cross entropy loss (NCE) can be be expressed as:

$$\mathcal{L}_{NCE} = \frac{\sum_k q(k|x) \log p(k|x)}{\sum_{k'} \sum_k q(k'|x) \log p(k|x)}, \tag{22}$$

where $q(k|x)$ is the ground truth probability that $x$ belongs to class $k$ and $p(k|x)$ is the predictive probability that $x$ belongs to class $k$.

**Asymmetric Generalized Cross Entropy(Classification)(Zhou et al., 2021)**: The Asymmetric Loss function is defined as the following formula:

$$\mathcal{L}^{\eta}_{\text{Asm}}(f(x), y) = (1 - \eta_x)\mathcal{L}(f(x), y) + \sum_{i \neq y} \eta_{x,i}\mathcal{L}(f(x), i) \tag{23}$$

where the $\mathcal{L}$ is the loss function need to be asymmetric, when we use the GCE (Zhang & Sabuncu, 2018) as the loss function, it would be the Asymmetric Generalized Cross Entropy Loss.

**LDR KL(Classification)Zhu et al. (2023)** The LDR-KL Loss is in the family of LDR losses based on KL divergence, the formula of the LDR-KL Loss is shown as followed:

$$\mathcal{L}_{\text{LDR-KL}}(f(x), y) = \lambda \log[\frac{1}{K} \sum_{k=1}^{K} (\frac{f_k(x) + c_{k,y} - f_y(x)}{\lambda})] \tag{24}$$

$\epsilon$ **-softmax(Classification) (Jialiang et al., 2024)** : The $\epsilon$ -softmax is a loss function which would strength the biggest value after softmax to help the prediction far away from classification boundary. The formula of the $\epsilon$ -softmax function would change the output as followed:

$$f'_k(x) = \begin{cases} \frac{f_k(x)}{m+1} & f_k(x) \neq argmax(f(x)) \\ \frac{m+f_k(x)}{m+1} & f_k(x) = argmax(f(x)) \end{cases} \tag{25}$$

where $m$ is a hyperparameter to control the output distance after the $\epsilon$- softmax. We use the $CE_\epsilon$ as the baseline which is also used in the paper (Jialiang et al., 2024).

## C  Hyperparameter Setting details

We will report the results of hyperparameters in this section, The parameter for the baselines are as followed: both $\alpha$ and $\beta$ are equal to 1 in RCE loss; q is 0.7 in Generalized Cross Entropy Loss and for NCE, we also use $\alpha$ and $\beta$ equals to 1 and 2.

## D  Compute Resources

We conduct all the experiments on a server which has two Intel Xeon 6326 CPU and 6 Nvidia A6000 GPU. We implement our code using the pytorch of version 2.4.1.

## E  Computational and model-size overhead

Table 6: Updated computational overhead summary on ResNet-18 (Batch size = 32, CPU). Following algorithmic optimization, KPD achieves inference latency and GFLOPs comparable to the baseline while maintaining a significant parameter reduction.

| Model Type | Rank | Patch | Params | Ratio | GFLOPs | Infer. (ms) |
|---|---|---|---|---|---|---|
| Normal (Baseline) | — | — | 11,181,642 | 1.000× | 0.074 | 7.3 |
| KPD | 100 | (4×4) | 2,562,042 | 0.229× | 0.076 | 7.9 |
| Rank Factorization | 100 | — | 2,549,442 | 0.228× | 0.076 | 7.1 |
| KPD | 100 | (2×2) | 2,625,642 | 0.235× | 0.076 | 8.0 |
| KPD | 100 | (8×8) | 2,514,042 | 0.225× | 0.076 | 7.1 |

Table 7: Updated ablation study on the Rank parameter in KPD (ResNet-18, Patch Size 4×4). The optimized implementation ensures that inference time remains low even as the rank increases.

| Model Type | Rank | Params | Ratio | GFLOPs | Infer. (ms) |
|---|---|---|---|---|---|
| Normal (Baseline) | — | 11,181,642 | 1.000× | 0.074 | 7.3 |
| KPD | 20 | 657,722 | 0.059× | 0.074 | 7.4 |
| KPD | 50 | 1,371,842 | 0.123× | 0.075 | 7.0 |
| KPD | 100 | 2,562,042 | 0.229× | 0.076 | 7.9 |
| KPD | 200 | 4,942,442 | 0.442× | 0.078 | 7.1 |

Table 8: Updated ablation study on the Patch Size in KPD (ResNet-18, Rank = 100). The optimized algorithm eliminates the previous latency discrepancies across different patch sizes.

| Model Type | Patch Size | Params | Ratio | GFLOPs | Infer. (ms) |
|---|---|---|---|---|---|
| Normal (Baseline) | — | 11,181,642 | 1.000× | 0.074 | 7.3 |
| KPD | (2×2) | 2,625,642 | 0.235× | 0.076 | 8.0 |
| KPD | (4×4) | 2,562,042 | 0.229× | 0.076 | 7.9 |
| KPD | (8×8) | 2,514,042 | 0.225× | 0.076 | 7.1 |

