# OpenReview forum: "Enhancing Model Robustness Against Noisy Labels via Kronecker Product Decomposition"
_TMLR — Rejected by TMLR_

### Review · Reviewer_U1Bx · 2025-12-31

**Summary Of Contributions:**

The paper proposes improving robustness to label noise by applying Kronecker product decomposition to model weight layers. Theoretically, using a result by Ma and Farrah, it shows that heavily overparameterized linear models can converge to suboptimal solutions by fitting label noise. Experiments on a toy dataset, FashionMNIST, CIFAR-10, and CIFAR-100 demonstrate that Kronecker product decompositions can improve performance under high label noise for several robust loss functions.

**Strengths:**
- The approach is compatible with any robust loss function
- The proposed Kronecker product decomposition can be straightforwardly applied to a wide range of network architectures.
- Overall, the paper is well written.

**Weaknesses:**
- The theoretical contribution is weak. (i) Its derivation is a simple application of a theorem by Ma and Farrah. (ii) The subsequent arguments rely on speculative statements such as “there is the possibility” or “there is a chance” that certain properties hold. The inability to extend the proof to the proposed setting does not strengthen the claim
 - The connection between theory and experiments is unclear. The theoretical assumptions are not reflected in the experimental setup (e.g., large overparameterization, the decomposed matrix functioning as the product matrix of a linear layer, and the use of different loss functions). As a result, the observed empirical improvements may be explained by simpler, well-understood mechanisms rather than the proposed theoretical insights (see below).

**Audience:**

Yes

**Audience Explanation:**

Improving the robustness of machine learning models is of broad interest to both the research community and practitioners working with real-world datasets.

**Broader Impact Concerns:**

No concerns

**Claims And Evidence:**

No

**Claims Explanation:**

While the paper presents the assumptions required for its theoretical result, it does not comment on the fact that these results rely on large overparameterization, an assumption that is not realized in the experiments. This questions, whether the theoretical analysis is meaningfully connected to the experimental results. In particular, the experimental results may be explained by a simple bias-variance trade off:
The experiments introduce substantial label noise (e.g., 55% in Section 5.1 and 40%–60% in Section 5.2). Although the total number of parameters is does not decrease by use of the Kronecker product decomposition, the rank of the resulting product matrix does, thereby lowering the number of effective parameters. This reduction in expressivity limits the ability to fit noisy labels—an effect that is well understood and sufficient to explain the observed performance gains.

**Requested Changes:**

**Major:**

- The experimental section should match the assumptions of the theoretical results.
- The experimental section should demonstrate that the observed improvements cannot be attributed to the well-known bias–variance trade-off, but instead reflect a distinct mechanism induced by the proposed method.

**Minor:**
- In 4.1, it says that “In this part, we make an assumption that eps_i = 0 with probability nu, and with probability of 1-nu, eps_i follows noise distribution P_0.“ This does not align with equation (4) and the rest of the paper uses (4) instead of this assumption.
Equation (5) is not technically correct (R_L is not defined by the arg min)
- Typo in 4.1 “problemequation“
- There are no formal Theorem 4.1 or Algorithm 5.2, which you refer to in 4.1 and 4.2
- 4.2 contains some formatting errors
- Please specify in 5.1 the range of the label function
- Typo in 5.1 “the the”

---

> ### Author Response · Authors · 2026-03-01
>
> We are very grateful for the time and effort the reviewers dedicated to providing such constructive and detailed feedback, which has been instrumental in strengthening our paper. Below, we provide our point-by-point responses to the specific concerns.
>
> **Weakness**
> 1. We agree that the result builds directly on the theorem of Ma & Farrah. Our contribution lies in applying this to reveal how non-convex factorization specifically alters the critical-point structure to improve robustness under label noise.
> 2. We theoretically demonstrate that such a possibility exists, and our experiments further verify that this phenomenon indeed occurs in practice. And we want to clarify that the first infimum means all possible \(A_j^\wedge\) and \(B_j^\wedge\) with any \(R\).
> We will use different symbols for these two \(R\)'s.
> The meaning of Equation 7 is that for some \(R\) (not all possible \(R\)), the decomposition of \(W^\*\) with rank \(R\) may not be identifiable.
> This implies that some members in \(\mathcal{S}\) are not identifiable, and some of them might be identifiable.
>
>
> **Requested Changes**:
>
> Major:
> 1. We are running additional tests with significantly larger ranks to better match the overparameterization assumptions of the theoretical results.
> 2. To show that KPD is not just a simple bias-variance trade-off from parameter reduction, our ablation study demonstrates that even when Rank R is increased (increasing expressivity), robustness gains are maintained or improved.
>
>
> Minor:
>
> 1. Thanks for pointing out, we corrected the Eq.5. And we checked the paper  and find that we are also using eta in section 4.1.
>
> 2. We fixed typos in Section 4.1 and 5.1 (e.g., "the the"). Algorithm 1 and Theorem 4.1 have been reformatted into proper environments.

---

> > ### Comment · Reviewer_U1Bx · 2026-03-02
> >
> > I thank the authors for their response and revisions.
> >
> > Regarding point (2), the authors refer to Table 3 from the ablation study, where a small table with three data points suggests that increasing the rank from 20 to 50 and 100 improves accuracy. However, these limited empirical results, together with the accompanying theoretical discussion, do not convincingly clarfiy the specific effects of KPD observed in the experiments.

---

> > > ### Author Response · Authors · 2026-03-14
> > >
> > > 1.We thank the reviewers for the suggestion. In the revised version, we have conducted additional experiments on the Clothing1M dataset and updated the results accordingly. Since Clothing1M is a large-scale real-world noisy dataset with different characteristics from CIFAR-100 and Tiny ImageNet, the hyperparameters of baseline methods (especially those with modified loss functions) need to be re-tuned to ensure a fair comparison. In the current revision, we additionally report results with two representative loss functions: cross‑entropy (CE) and $\epsilon$‑softmax. Cross-entropy (CE) is included as a standard benchmark, while $\epsilon$-softmax is a recently proposed robust loss and serves as a representative modern baseline. The results show that our method consistently outperforms the corresponding baseline models under both loss functions, demonstrating that the proposed approach remains effective on real-world noisy datasets. We will continue to update the experimental results for the remaining loss functions. Due to the limited rebuttal time, we report the results for two representative loss functions here, and the same protocol will be applied to the remaining losses in the final version.
> > >
> > > 2.We thank the reviewer for the suggestion. In the revised version, we further extend the ablation study in Table 4 by including additional experiments with larger ranks ($R=150$ and $R=200$). The new results show that although the accuracy initially improves as the rank increases, the performance starts to degrade when the rank becomes too large, indicating that excessively large rank makes the optimization more difficult and leads to worse generalization. This observation is also aligned with our theoretical analysis in Section 4.1. According to Theorem 4.1, under label noise, the ground-truth weight matrix $W^*$ is not necessarily the unique minimizer of the noisy empirical risk. As the rank $R$ increases, the hypothesis space induced by KPD becomes larger, giving the model more degrees of freedom to fit noisy labels.

---

### Review · Reviewer_CLjZ · 2026-02-05

**Summary Of Contributions:**

This paper proposes using Kronecker Product Decomposition (KPD) as an in-training modification to improve robustness to noisy labels/outputs. The main idea is to replace each weight matrix with a sum of Kronecker products during optimization, yielding a non-convex training problem and a corresponding training procedure. The authors claim the approach is easy to combine with robust loss functions and empirically improves performance over robust-loss baselines.

**Audience:**

Yes

**Audience Explanation:**

Despite the weaknesses, the paper may interest readers studying robustness to label noise and the interaction between optimization landscape / parameterization and generalization. The approach frames KPD as a plug-in parameterization that can be combined with robust losses (rather than proposing yet another loss). The ablation discussion also connects robustness to model parameter size via KPD hyperparameters (rank and patch size), which touches on broader questions about capacity and robustness. As an idea, “robustness via structured re-parameterization” could be useful if validated at scale.

**Broader Impact Concerns:**

No ethical concerns.

**Claims And Evidence:**

No

**Claims Explanation:**

Several central claims are only weakly supported. Empirically, the evaluation is limited to a small synthetic regression setup (200 samples total, 100 train / 100 test) and CIFAR/FashionMNIST-scale classification datasets. This makes it hard to conclude that the method generalizes to realistic, large-scale settings typical for TMLR readers. Additionally, although the paper notes KPD can increase or decrease the number of parameters, it provides no clear accounting of the added training/inference cost.

**Requested Changes:**

A. Writing / presentation fixes (blocking issues).

* Please correct obvious writing and formatting problems that reduce readability and trust. For example, in Section 4.2 there is a line consisting of a single bracketed citation marker (“[1]”) followed immediately by pseudo-code that is not typeset as an algorithm environment. This should be properly formatted (caption, inputs/outputs, numbered steps) and the stray citation marker should be removed or integrated correctly.

* Do a full proofreading pass (there are visible typos such as “imrove robustness”).


B. Strengthen empirical evidence beyond CIFAR-level.

* Expand experiments to larger-scale datasets and more realistic noise (e.g., WebVision-style noise or large-scale curated datasets with synthetic noise). Current classification results are limited to FashionMNIST/CIFAR-10/CIFAR-100.

* The regression evidence is currently based on a very small synthetic setup (200 samples). Add real-world regression datasets and larger sample sizes, and report confidence intervals/significance where appropriate.


C. Report computational and model-size overhead clearly.

* Since KPD can change parameter count, you should report parameter counts, FLOPs, memory, and wall-clock training/inference time for Normal vs KPD vs rank factorization.

* Discuss the practical trade-off: robustness gains vs added computation, and how this varies with rank/patch size (especially because you already argue these hyperparameters affect parameter size).


D. Clarify scope of theoretical claims.

Make the limitations explicit: the theory focuses on simplified linear settings, while the headline claims target deep models. Better align claims/conclusions with what is actually proven and tested.

---

> ### Author Response · Authors · 2026-03-01
>
> We sincerely appreciate the insightful comments, we have carefully considered all the feedback and updated the manuscript. Our detailed responses to the specific concerns are as follows.
>
> **Requested Changes**
>
> A. We have reformatted Algorithm 1 (formerly 5.2) and Theorem 4.1 using proper LaTeX environments. We also performed a full proofreading pass to remove stray citation markers and correct typos.
>
> B. We thank the reviewer for this constructive feedback. We recognize that validating our method on larger-scale datasets is essential. We are currently conducting experiments on the Clothing1M dataset to demonstrate generalizability to realistic, instance-dependent noise.
>
> C. We have added an extra section in the Appendix 5 to report the computational and model-size overhead.
> We also include a summarized table here:
> | Model Type | Rank | Patch  | Params  | Ratio |
> |:-------|:------:|:------:|:------:|-------:|
> | Normal  | - | - | 11M | 1.000x |
> | Rank Factorization | 100 | - | 2.54M | 0.228x |
> | KPD | 100 | (2,2) | 2.62M | 0.235x|
> | KPD | 100 | (4,4) | 2.56M | 0.229x|
> | KPD | 100 | (8,8) | 2.51M | 0.225x|
> | KPD | 20 | (4,4) | 0.65M | 0.059x|
> | KPD | 50 | (4,4) | 1.37M | 0.123x|
> | KPD | 200 | (4,4) | 4.94M | 0.442x|
> | KPD | 100 | (2,2) | 2.62M | 0.235x|
>
> D. We include this in section 4.1 to clarify: “It is important to note that while Theorem 4.1 focuses on the linear setting to provide a tractable analysis of identifiability under noise, it serves primarily as a conceptual motivation for deep models.”

---

### Review · Reviewer_sh7F · 2026-02-13

**Summary Of Contributions:**

This paper studies the problem of training neural networks under noisy labels and proposes to enhance robustness by replacing each weight matrix with a Kronecker Product Decomposition (KPD). The authors first present a theoretical result showing that, for a single-layer linear regression model trained with $L_1$ loss under label noise, the ground-truth weight matrix $W^*$ may not be identifiable. Motivated by this, they propose to introduce non-convexity via KPD and argue that this may allow optimization to converge to better local minima. The method is evaluated on synthetic benchmarks under synthetic symmetric and asymmetric noise. Empirically, the proposed approach often improves performance compared to standard training.

===========

**Strengths**

1.	The proposed method can be combined with existing robust loss functions.

2.	The classification experiments cover multiple datasets, noise types (symmetric and asymmetric), and loss functions.

==========

**Weaknesses and Concerns**

1. The theoretical result in Section 4.1 shows that, under specific assumptions (single-layer linear regression, Gaussian inputs, $L_1$ loss, particular noise model, and $N \le 0.1m$), the ground-truth $W^*$ may not be a local minimum of the empirical risk. The provided theoretical analysis seems trivial. In the presence of label noise, minimizing empirical risk over noisy data need not recover the clean ground-truth parameter.

2. The theoretical section has other sigificant limitations:
	•	The result does not extend to deep nonlinear networks.
	•	There is no formal analysis showing that KPD restores identifiability or guarantees improved robustness.
	•	The argument that non-convexity may help is speculative and not rigorously justified.


3. It remains unclear why KPD improves robustness. Is the effect due to structured parameterization? Implicit regularization? Low-rank constraints? Simply increased model capacity?

4. The method introduces additional hyperparameters (rank, patch size, layer-wise configurations) without principled selection guidelines. Performance appears sensitive to these choices. This substantially increases tuning complexity compared to standard training or robust loss functions.

5. All classification experiments rely on synthetic symmetric or asymmetric noise. There is no evaluation on real-world noisy datasets (e.g., Clothing1M, WebVision), where noise is instance-dependent and more challenging. Furthermore, comparisons are limited to robust loss functions and do not include stronger noisy-label training frameworks (e.g., DivideMix, Co-teaching).

**Audience:**

Yes

**Audience Explanation:**

Researchers in robust learning under label noise, implicit regularization, and structured matrix factorization would likely be interested.

**Claims And Evidence:**

No

**Claims Explanation:**

The claims made in the submission are not sufficiently supported by accurate, convincing, and clear evidence for the following reasons:

1.	The theory does not demonstrate that KPD restores identifiability or guarantees improved robustness; it merely motivates the introduction of non-convexity. As such, the theoretical section does not support the robustness claims made for deep models.

2.	The paper does not clearly establish why KPD improves robustness. It remains unclear whether the gains arise from structured factorization, implicit regularization, low-rank constraints, or simply increased model capacity.

3.	All classification experiments are conducted under artificially injected symmetric or asymmetric noise. The paper does not evaluate on real-world noisy-label datasets, where noise is more complex and instance-dependent.

4.  The comparisons focus mainly on different robust loss functions. The method is not compared against stronger noisy-label learning frameworks (e.g., semi-supervised or co-training approaches), which weakens the claim that the proposed approach provides meaningful advancement over existing techniques.

**Requested Changes:**

1.	Strengthen the theoretical analysis.
* Clarify the precise role of KPD in improving robustness beyond introducing non-convexity.
* Provide a deeper analysis (or at least stronger intuition) connecting KPD to robustness in deep nonlinear networks.
* If formal guarantees are not feasible, clearly state the limitations of the current theoretical result and avoid overclaiming.


2.	Evaluate on real-world noisy datasets.
* Add experiments on at least one real noisy-label benchmark (e.g., Clothing1M or WebVision).
* Discuss how the method performs under instance-dependent and realistic noise patterns.

3.	Compare with stronger noisy-label learning methods.
* Include comparisons with more advanced approaches such as co-training, semi-supervised, or noise-adaptation methods.
* Position the method clearly relative to the current state of the art.

4.	Clarify hyperparameter selection and practical considerations.
* Provide principled guidelines or heuristics for choosing rank and decomposition shape.
* Discuss computational cost, memory overhead, and training stability.
* Analyze sensitivity to rank and patch size more systematically.

5. Theorem 4.1 is referenced in the main text, but its formal statement (with theorem numbering) seems missing.

---

> ### Author Response · Authors · 2026-03-01
>
> We thank the reviewers for their time and the constructive feedback provided, which has helped us significantly improve the quality and clarity of our manuscript. Below, we address each of comments point by point:
>
> **Weakness and concerns**
> 1. The inability of standard ERM to recover ground-truth parameters under noise is precisely the problem our work addresses. Theorem 4.1 proves that the true optimal $W^*$ is not a local minimum in noisy, convex ERM landscapes. By introducing KPD, we transition to a non-convex landscape. This structural re-parameterization enables the optimization to bypass noise-induced minima and converge closer to the true optimal solution, which is inherently more robust to diverse noise patterns.
> 2.  Since theorem 4.1 shows that noisy convex landescapt leads to a non-optimal solution, we get the intuition that introducing non-convexity would help us to find a better solution which is confirmed by our extensive empirical study.
> 3. We believe  KPD improves robustness primarily through implicit regularization by restricting the parameter search space. By imposing a structured Kronecker bottleneck, it limits the model's effective degrees of freedom. This constrained search space prevents the network from easily memorizing unstructured random label noise, a process that typically requires dense, unconstrained parameterization. Consequently, KPD significantly reduces the risk of overfitting to corrupted labels, allowing the model to better capture the underlying clean distribution.
> 4. Our ablation study already indicates that higher rank generally improves robustness, and we will provide heuristics for selecting patch sizes based on layer dimensions.
> 5. We want to emphasize that KPD is an in-training approach and does not need any architecture change during inference. Therefore, we picked the baselines belonging to in-training methods. Any method that introduces an architecture change can be applied on top of our method to get even better performance
>
> **Requested Changes:**
> 1. We theoretically demonstrate that such a possibility exists, and our experiments further verify that this phenomenon indeed occurs in practice. And we want to clarify that the first infimum means all possible $\hat A_j$ and $\hat B_j$ with any $R$.
> The meaning of Equation 7 is that for some $R$ (not all possible $R$), the decomposition of $W^*$ with rank $R$ may not be identifiable. This implies that some members in $\mathcal{S}$ are not identifiable, and some of them may be identifiable.
> 2. We are currently conducting additional experiments on real-world noisy datasets and integrating KPD into more advanced training frameworks. We will include these comparative results in the revised manuscript.
> 3. We are running experiments with stronger noisy-label learning methods and would update when we have results. We want to emphasise that our method is particularly designed for in-training methods and is orthogonal to the most robust research requiring to change the model’ architecture. So we did a diverse and comprehensive experiment with robust loss functions existed in the literature.
> 4. We have added detailed tables in the Appendix 5 to illustrate computational cost, including parameter counts, inference time and GFLOPs. We want to point out that since after training, the model weight could be re-calculated to a matrix with the original shape, the inference time would be similar as the original model. We also include a summarized table here:
> | Model Type | Rank | Patch  | Params  | Ratio |
> |:-------|:------:|:------:|:------:|-------:|
> | Normal  | - | - | 11M | 1.000x |
> | Rank Factorization | 100 | - | 2.54M | 0.228x |
> | KPD | 100 | (2,2) | 2.62M | 0.235x|
> | KPD | 100 | (4,4) | 2.56M | 0.229x|
> | KPD | 100 | (8,8) | 2.51M | 0.225x|
> | KPD | 20 | (4,4) | 0.65M | 0.059x|
> | KPD | 50 | (4,4) | 1.37M | 0.123x|
> | KPD | 200 | (4,4) | 4.94M | 0.442x|
> | KPD | 100 | (2,2) | 2.62M | 0.235x|
> 5. Theorem 4.1 has been reformatted using a proper LaTeX environment for better clarity.

---

### Decision · Action_Editor_NP9x · 2026-04-02

**Recommendation:** Reject

**Audience:**

Yes

**Audience Explanation:**

Researchers in robust learning under label noise, implicit regularization, and structured matrix factorization would likely be interested in the findings of this paper. More broadly, improving the robustness of machine learning models is of broad interest to both the research community and practitioners working with real-world datasets.

**Claims And Evidence:**

No

**Claims Explanation:**

All reviewers raise concerns that a) the theoretical results are derived under strong assumptions that are not matched by the empirical results, b) the empirical results themselves are provided on small-scale datasets that are not reflective of large-scale situations. All the reviewers stated that the authors' response did not address these fundamental concerns.